# A Narrative Review of the History of Burn-Related Depression and Stress Reactions

**DOI:** 10.3390/medicina58101395

**Published:** 2022-10-05

**Authors:** Shivan N. Chokshi, Carter M. Powell, Yulia Gavrilova, Steven E. Wolf, Deepak K. Ozhathil

**Affiliations:** 1Department of Surgery, University of Texas Medical Branch at Galveston, Galveston, TX 77550, USA; 2Department of Surgery, Medical University of South Carolina, Charleston, SC 29464, USA

**Keywords:** depressions, burns, ASD, stress reaction, PTSD, thermal injury, history

## Abstract

While the roots of burn care date back several millennia, recognition and treatment of psychiatric trauma has had a more contemporary journey. Our understanding of burn care has evolved largely separately from our understanding of psychiatry; however, proper care of the burn patient relies on the comprehension of both disciplines. Historically, high burn mortality rates have caused clinicians to focus on the physiological causes of burn mortality. As burn care improved in the 20th century, providers began to focus on the long-term health outcomes of burn patients, including mitigating mental health consequences of trauma. This shift coincided with advances in our understanding of psychological sequelae of trauma. Subsequently, an association between burn trauma and mental illness began to emerge. The current standard of care is the result of thousands of years of evolving practices and theories, yet our understanding of the pathophysiology of depression among survivors of severe burn injury is far from complete. By taking measure of the past, we aim to provide context and evidence for our current standards and emphasize areas for future lines of research.

## 1. Introduction

Burn-related trauma can be devastating and life-altering. According to the National Center for Injury Prevention and Control, thermal burns caused by fires remain one of the leading causes of preventable injury and the seventh most common cause of death among young children in the United States [1]. Approximately 350,000 to 450,000 patients present to hospitals and clinics around the country every year to have burn injuries evaluated. Over 40,000 of these warrant hospitalization and despite remarkable advances in burn care, there remains a 3% overall mortality rate among patients admitted to specialized burn centers [2].

While most burn injuries are not life-threatening and do not result in significant aesthetic or physiological changes, severe burn injuries can have long-lasting effects on function and identity. Hypertrophic scar formation and subsequent scar contractures are common drivers of burn-related morbidity and loss of function. Hypertrophic scars are defined by the proliferation of dermal tissue that result in visible and elevated scars following an injury to the skin. These scars can persist for long periods of time after acute burn care is completed and can significantly alter a patient’s appearance. Scar contracture can cause devastating functional problems such as limited range of motion, deformity and disability [3]. If deeper tissues are affected, joints can become subluxed or dislocated with tightening of ligaments. The limited range of motion caused by joint contracture is associated with impaired physical functioning, hindering patients’ return to employment and reintegration into society [4,5]. Contractures in the mouth and neck may cause mandibular deformity and loss of normal dental occlusion. Burn-related microstomia and neck contracture can lead to impaired communication and reduced food intake [6,7]. In addition to the devastating physical impairments associated with burn-related contracture, some patients may even require amputation [8]. Despite innovations in prosthetics, amputation may lead to a host of complications like muscle weakness, skin breakdown and phantom limb pain [9]. Hypertrophic scarring, joint subluxation, maxillofacial deformities and amputations cause obvious impairments in function, but they also cause life altering changes to identity and self-image. The psychiatric impact of such conditions has not been well studied because burn providers have historically focused on objective outcomes of severe burns, like mortality and morbidity. Even with improving mortality and morbidity, collecting objective data remains challenging.

Until recently, little attention has been devoted to the psychological outcomes of burn trauma survivors. An awareness of trauma-related psychological complications has existed since the ancient text of *The Illiad* by Homer, which detailed soldiers traumatized by the adversities of war, such as the intense grief felt by Achilles upon the death of his closest friend Patroclus [10,11]. This awareness evolved slowly, lacking the specific language and diagnoses necessary to describe and differentiate the psychological effects of trauma. However, great strides have been made in the field of psychology in the last century. Recognition of depressive reactions and the inclusion of specific diagnoses, such as Posttraumatic Stress Disorder (PTSD) and Acute Stress Disorder (ASD) by the Diagnostic and Statistical Manual (DSM) of Mental Disorders-III and DSM-IV, respectively, have improved our understanding of both normal and pathological stress reactions following trauma [12,13].

Only recently has the understanding of trauma-related psychological complications extended to burn care. Burn trauma has been found to result in several psychopathologic effects on survivors, ranging from ASD, which occurs within the first 30 days following a traumatic event, PTSD, which notes a stress reaction that persists beyond 30 days from an incident, to major depressive disorder, which requires at least five diagnositic criteria to be present for a two-week period [14]. Systematic reviews report that up to 30% of burn survivors develop ASD, up to 45% develop PTSD and up to 54% develop at least “mild” depressive symptoms [15,16]. This burn-related psychological trauma subsequently impairs physical function and quality of life. Recognizing the importance of ASD and PTSD, the American Burn Association (ABA) recommended in 2012 that all burn patients receive screening for depression, ASD and PTSD [17]. In 2019, the ABA began requiring psychological screening for PTSD, depression, suicidal ideation and substance abuse and relevant intervention for burn centers as an aspect of accreditation [18]. Burn survivors with psychiatric disorders are likely to experience longer hospital length of stay, more surgical procedures, higher levels of dysfunction, an increased need for medical assistance, and an increased risk of suicide [19,20,21]. Two studies reported that suicide rates in burn survivors are nearly five times higher than in the general population [19,22]. Other studies have measured suicidal ideation in burn patients, finding a positive correlation between acute pain at discharge and suicidal ideation [23]. Despite increased efforts to study post-burn psychological effects, the paucity and heterogeneity of data have made quantifying psychiatric illness difficult in burn patients [19,21]. While the extent of these associations is unclear, a better understanding of the psychological sequelae of burn patients and appropriate screening of at-risk individuals has the potential to save lives.

Though previously overshadowed by the urgency of acute wound management, treating concurrent psychological conditions has become an integral part of modern burn care. Common challenges with activities of daily living can lead to psychological distress among survivors of burn trauma. Patients are often excited to go home, but they may not anticipate the subsequent challenges of a previously familiar environment in the context of new physical and psychological limitations. They may underestimate the unwanted attention from strangers. Previously enjoyable foods may be challenging to eat. Family members can fail to fully anticipate the amount of support the patient may need. In the absence of physical limitations, burn patients may still experience psychological trauma related to the accident, their appearance, or persistent pain [24]. Avoidance of trauma and injury reminders (e.g., thoughts, memories, people, conversations, places, objects, activities) is a hallmark symptom of and risk factor for PTSD. Such behaviors may significantly limit the survivor’s psychosocial functionality and can interfere with their recovery process [25,26]. Substance use, withdrawal from social activities, and other maladaptive behaviors have also been linked to avoidant coping style following trauma [27,28]. The realization that life may not return to a pre-injury sense of normal can have a significant emotional impact, resulting in profound negative reactions such as feelings of helplessness, hopelessness and loneliness. Sequelae of such degrees of emotional stress include difficulty falling asleep, substance abuse and suicidality [29].

As caregivers, we must ensure patients are given adequate emotional support following a burn injury to transition back to normal life. Currently, several resources are available to burn survivors, such as the Phoenix Society for Burn Survivors (https://www.phoenix-society.org), which may help patients adapt to their new lifestyle. However, the presentation and availability of these resources can vary among burn centers, potentially leaving the psychosocial needs of some patients unmet. Data from a population of trauma victims including burn patients, suggest nearly half of injured patients in the post-acute care setting did not know how or where to get psychological help after injury [30]. While an initial focus on physical limitations may be necessary to ease pain and regain function, comprehensive treatment must also address psychiatric trauma, which has the potential to significantly outlast most physical limitations.

The purpose of this paper is to explore the relationship between the evolution of modern burn care and our understanding and management of mental health disorders, specifically depression, in burn trauma survivors. By recording the history of burn-related depression with such a broad scope, we hope to fill the gaps in the current literature and provide context and evidence for current practice patterns in burn care. We will delve into current mental health practices and explore the alternative therapies that preceded it. To fully appreciate the context of current burn care, it is crucial to investigate two converging timelines. The first is the general advancement of burn wound care. The second are the breakthroughs in our understanding of depression and acute stress reactions, both in its neuropathologic mechanism as well as in the psychosocial severity of its impact. In short, this paper tells the story of the confluence of burn care and burn-related psychiatric reactions from its inception in ancient times to the modern standard of care. The confluence of these two trends sets the stage for future advances in the treatment of burn-related depression and acute stress reactions. Furthermore this narrative seeks to identify holistic topics on the horizon of psychiatric advancement that would be of interest to burn care providors as well as specific suggestions for research to improve the field. We hope that by exploring the origins of burn-related depression and acute stress reaction management, future investigators will be able to more effectively influence the trajectory of the field.

## 2. Ancient History

The discovery and control of fire has had a major impact on the course of human evolution through its offering of light and heat; however, its introduction also normalized the risk of thermal injury and the subsequent need for burn wound treatment [31]. Burn care was limited in the ancient world with much of medical doctrine involving assumptions about supernatural forces and their role in wound healing. The earliest records on Egyptian scrolls from around 1500 BCE describe burn dressings using milk from mothers, accompanied by ceremonies calling upon the Goddess Isis, the goddess of life and magic [32].

With little knowledge at the time of the pathophysiology of burns, healers looked to products that were familiar to them. Ancient Egyptians focused on thyme, opium and belladonna for pain relief. Ancient Greek and Roman accounts identified the use of pig fat, resin and bitumen for burn treatment. Ayurvedic (ancient Indian) medical texts regularly included honey in burn dressing and healing ointments. Similarly, burn treatment in traditional Chinese medicine was a combination of philosophy, knowledge and herbal medicine [32]. While many of these remedies provided a mild short-term alleviation of pain, people with more severe burns did not experience significant benefits. Indeed, patients with severe burns could only languish in hospitals while their infected skin sloughed off, leaving open wounds that led to contractures and life-long disabilities or worse, sepsis and death. [32]. With such poor outcomes, the psychological health and quality of life of patients were of little concern to physicians at that time. Survival itself was a huge success.

Due to the high mortality of burn injury and lack of scientific reasoning, there was no known association between burns and psychological disorders like depression. In antiquity, there was little fundamental understanding of psychology; however, preliminary ideas were being formed about the existence of a suboptimal behavioral condition. In the second millennium BCE, the Babylonians in Mesopotamia provided the earliest known records of what we now understand to be depression. King Hammurabi, the first Babylonian king, described behavioral and psychological changes in early medical texts. These changes were dominated by observable reactions. Among these Babylonian descriptors were “*asustu*” (distress), “*zikurrudu*” (suicidal) and “*hip libbi*” (breaking of the mind), which were generally followed by a ritualistic treatment [33]. Translated early Babylonian texts affored an important window into our early understanding of depression related behavioral changes:

“*If an awīlum [or specifically the head of a household] has had a [long] spell of misfortune—and he does not know how it came upon him—so that he has continually suffered losses and deprivation (including) losses of barley and silver and losses of slaves and slave-girls, and there have been cases of oxen, horses, sheep, dogs and pigs, and even [other] awīlū [in his household] dying off altogether; if he has frequent nervous breakdowns, and from constantly giving orders with no [one] complying, calling with no [one] answering, and striving to achieve his desires while having [at the same time] to look after his household, he shakes with fear in his bedroom and his limbs have become “weak”; if because of his condition he is filled with anger against god and king; if his limbs often hang limp, and he is sometimes so frightened that he cannot sleep by day or night and constantly sees disturbing dreams; if he has a “weakness” in his limbs [from] not having enough food and drink; and if [in speech] he forgets [cannot find] the word which he is trying to say; then, as for that awīlum, the anger of [his] god and goddess is upon him*”.

These descriptions suggest two important findings. First, there was an emphasis on observations in lieu of discussing subjective thoughts or feelings. The author observes fear, rejection and even weakness, but makes no conclusion on the impact these have on the individuals’ psychology [33]. The Babylonians recognized these behaviors as abnormal but did not know what labels or terms could be used to describe them. Second, it is important to note the attribution of all the maladies to the anger of a god or goddess, highlighting the belief that depression was mediated by supernatural and divine forces.

Individuals with mental illnesses were poorly understood and their afflictions were thought to be caused by demonic possession, emotional deities or witchcraft [34]. As a result, treatments were handled by priests, rather than physicians [33]. Often, the skulls of those with mental illnesses underwent trephination—a procedure in which a hole was made in the cranium using stone instruments to evacuate evil spirits. Various other “treatments” such as beatings, exorcisms, starvation and imprisonment were utilized on those with mental illnesses [35,36]. These inhumane treatments likely discouraged most individuals from disclosing their condition and provide context for the relatively limited records of depression among trauma victims, including burn survivors.

Perhaps the most significant advancement in the ancient understanding of depression was made by Hippocrates (460-370 BCE), who argued that depression was rooted in abnormal bodily functions, rather than supernatural influances. Hippocrates proposed that all bodily mechanisms were regulated by four internal fluids referred to as humors: blood, black bile, yellow bile and phlegm. He attributed the melancholic state—what we now understand as depression—to an excess of black bile, which was characterized by fear and sadness [37]. His ideas were echoed by the Roman philosopher Galen (129-210 CE), who expanded on the set of symptoms that defined melancholy to also include delusions. The treatments of choice proposed by these two philosophers included bloodletting, baths and exercise. Interestingly, Galen also commented on wound care, noting that wounds healed optimally in moist environment, thus providing a framework for detoxifying burn wounds with hot oil [38]. As a result, the Galenic practice of re-burning burn wounds with hot oil became dogma for more than a millennia. It remained the standard of care until Ambrose Paré challenged it in 1535, establishing the roots of the modern practice of evidence based medical treatment.

Cornelius Celsus (25 BCE—50 CE), another Roman academic, was also an important figure in the early categorization of depression. His encyclopedic treatise *De Medicina* was among the best sources of knowledge on medicine in the Roman world. Despite the breadth of his knowledge, even he advocated the harsh treatments for depression that were accepted at the time, including binding with shackles, starvation and beatings [39]. Though each of these philosophers—Hippocrates, Galen and Celsus—held varying opinions on the origins and optimal treatments for mental illness, they were bound together by a relative lack of neurological understanding and a general apathy towards the individuals affected.

The understanding of burn care and depression in ancient times was severely limited by a preference for supernatural explanations, high mortality rates and a lack of tools necessary to discover the mechanisms of each. Patients fortunate enough to survive the initial trauma of their burn injuries, often suffered long-term physiological and psychological wounds due to misunderstandings that prevented ancient caregivers from properly addressing their psychiatric needs.

## 3. Post-Classical History and the Middle Ages (5th–17th Century)

Despite the technological revolutions and globalization of the post-Classical period, the heavy influence of religion and a reliance on classical teachings limited scientific advancement. As a result, little ground was gained in the understanding of burn care and psychological disorders. Early medieval medical practices in Western Europe were based on surviving Greek and Roman teachings. Furthermore, early medical writings (700-1100 CE) were disorganized, interwoven heavily with folklore and conceptually nebulous [40].

Modern scholarship has scant information to reconstruct the experiences of burn victims during the middle ages [41]. Historical accounts of fires and burn injuries from the medieval period focused on the urban environment and the risk factors that precipitated the fire, rather than the people that were marred by them. Of the records that do exist, little is mentioned regarding the effects of their injury. One possible reason for this was that significant burn trauma nearly always resulted in death, while milder burns typically healed well enough to not warrant further discussion.

Although the ancient Greeks paved the way for mental illness to be considered an organic problem rather than a supernatural one, the Black Plague in Afro-Eurasia brought a period of profound academic regression. The Black Plague in the mid-14th century resulted in the death of 70–200 million people and remains the most fatal pandemic in human history [42]. The widespread devastation, combined with the increasing influence of Christianity on European thinking, resurrected the idea that diseases originated from supernatural and malignant spirits.

In addition, asylums began to appear around Europe in which people showing symptoms of mental illness were imprisoned [35]. The asylums were not facilities intended for rehabilitation, but were institutions where the mentally ill were abandoned by their families to face inhumane treatment [35]. At La Bicetre, a typical asylum hospital in Paris, patients were shackled to the walls with iron cuffs, restricting all but the smallest movement and forcing them to sleep standing up. The rooms were never cleaned and patients were often left to live in their excrements [35]. Treatment had not advanced much since the days of Hippocrates, revolving around bloodletting, purging and ice baths. Religious and supernatural explanations persisted during the Renaissance, when the blame of tragedies fell on witches and diabolical possession. The accusatory nature of depression in Medieval Europe caused those afflicted to hold intense feelings of guilt and sinfulness, many of which were imagined [43]. The shadow of this legacy extends for centuries after, with social stigmatization of mental illness impairing academic curiosity and investments in psychiactric research well into the 20th century.

An important shift began in the 12th century, when medieval medicine, particularly that of Western Europe, became institutionalized within universities, which had established durable methods of formal record keeping [44]. Perhaps as a result of such records, the treatment strategies for burns began to shift. Building upon the principles of Galen, burns were prevented from drying out by using various ointments of vinegar, rose oil, egg and herbs [45]. The ointments were applied to the site of injury and reapplied if necessary. It is unclear whether these changes truly improved patient outcomes, but the shift in care exemplified a growing interest and understanding of the pathophysiology of burn care. Although the primary goal of these advances was a reduction in overall mortality, such changes also elucidated the secondary manifestations of burn injury, including its emotional sequelae. Unfortunately, the understanding of mental illness at the time had not concurrently developed to a state to provide any psychological support to the burn victims fortunate enough to survive their injuries.

Late medieval medicine was characterized by formal bodies of work that attributed disease to natural causes. While sin was involved in the general body of belief, it was only connected more generally via claims that diseases were a manifestation of humanity’s fallen state from the divine perfection of God. Importantly, sickness was understood to have originated from natural causes [46]. An examination of 57 descriptions of mental illnesses from pre-crusade chronicles and saints’ lives found that only a small proportion of these descriptions attributed mental illness to sin or wrongdoing, and in those cases the authors were using this attribution for religious propaganda [46]. Most of the reviewed texts indicated that authors were aware of Hippocrates’ argument that mental illness was caused by factors of humoral imbalance, overwork and grief [46].

Perhaps most importantly, the 16th and 17th centuries were marked by revolutions in scientific thinking. Religious ideology made way for natural philosophy and deductive reasoning. With regard to burn care, Ambroise Paré (1510–1590) brought the first advancement in empirical thinking in 1535, when he ran out of boiling oil, the standard treatment of burn care, while on a military campaign for Napoleon and was forced to use a balm of egg yolks, rose oil and turpentine [47]. Taking advantage of his impromptu natural experiment, Paré noted that those treated with the balm experienced better outcomes and subsequently adjusted his treatment protocols. Not only did Paré alter the course of burn management, he helped introduce the concept of evidence-based medicine. His contemporaries were similarly interested in scientific reasoning. During this period, Galileo, Leonardo da Vinci and William Harvey were, respectively, challenging perceptions on the universe, anatomy and the circulatory system using their own evidence-based practices [48]. The empirical and deductive reasoning that originated in this period was still limited by the tools and technology available, but it provided the future framework to understand the relationship between burns and depression.

## 4. Modern History (1700–1950)

### 4.1. Improvements in Burn Care

The 18th and 19th centuries were a period of remarkable progress in burn care and understanding of burn pathophysiology. A number of advances in surgical care including William Green Morton’s introduction of general anesthesia as well as Louis Pasteur and Joseph Lister’s work on microbiology and antiseptic techniques helped improve surgical outcomes. Systems of classification, such as the total body surface area (TBSA) to measure size of injury and burn degree to describe depth of injury were introduced. These led to TBSA-dependent fluid resuscitation in early burn management. In the early 20th century, Francis D. Moore’s observations of respiratory infections and pulmonary edema in victims of the Cocoanut Grove tragedy culminated in the first formula for fluid therapy in burn patients based on TBSA [49,50]. In 1928, Sir Alexander Fleming discovered penicillin, which proved to have an enormous impact in the fight against the microbial causes of sepsis that plagued burn patients. These advancements in acute care allowed physicians and researchers to turn their attention to the more subtle effects of burn injury, particularly psychological sequelae that had been overshadowed by high mortality rates.

### 4.2. Improvements in Psychiatry

While the management of burn wounds underwent rapid advancement during this time period, knowledge about depression had yet to gain similar academic momentum. Physicians lacked the tools to make specific diagnoses due to a limited understanding of psychiatry and consequently a lack of disease-targeted treatments, yet the number of “insane” patients in the 19th century grew rapidly [51]. Asylums were heavily utilized, likely due to the prevalence of neurosyphilis, and without proper diagnoses and treatments, they became the treatment of choice for a variety of psychological and psychiatric afflictions [52]. Society still misunderstood and mistrusted mental illness. Psychiatry was largely still outside the mainstream of science and psychiatric illnesses were clouded with uncertainty.

Well into the 20th century, psychiatric innovation was still in its infancy. In the 1940s, when other industries were pioneering jet engines and computers, medicine awarded the Nobel Prize to António Caetano de Abreu Freire Egas Moniz (i.e., Egas Moniz), a Portuguese neurologist who introduced the use of lobotomies to treat psychiatric disorders like schizophrenia, anxiety and depression [53]. The lobotomy (called prefrontal leucotomy at the time) is now considered a medical misdoing, having caused irreparable harm in the name of medicine. It highlights the primitive nature of psychiatry at the time and exemplifies the lack of understanding surrounding complex mental illness.

Despite these shortcomings, psychiatry did make notable progress during this era. In 1927, Julius Wagner-Jauregg developed a malarial therapy for neurosyphilis, which was linked to a form of psychosis characterized by grand delusions, paralysis and dementia. Before 1945, 5–10% of all psychiatric patient admissions were attributed to neurosyphilis, making Wagner-Jauregg’s malarial therapy a key treatment for a devastating disease [54]. Concurrently, electroconvulsive therapy (ECT) emerged as a safer and more effective intervention for treatment of depression, mania and catatonia [55]. Effective treatments gave more meaning to specific diagnoses, which require a more precise understanding of the relationship between the etiology and symptomology of a disorder.

The geopolitical conflicts of the 20th century allowed ample opportunity to observe the psychological ramifications associated with trauma. In 1915, Charles Myers used the term “shell shock” to describe the panic and sleep troubles experienced by soldiers in World War I, some of whom never experienced direct hand-to-hand combat [10]. In 1948, following the devastation of World War II (WWII), the World Health Organization included “acute situational adjustment” in their sixth version of the International Classification of Disease (ICD-6), followed by the introduction of “gross stress reaction” and “depressive reaction” in the American Psychiatric Association’s DSM-I [56,57]. These wartime observations of transient stress reactions would help prime the field to recognize similar associations between trauma and psychological disorders among civilians.

By the end of WWII, psychiatry had achieved a three-fold effect. First, an improved understanding of psychiatric illness spurred by psychological and pharmacologic advances in treatments, as well as an increased societal awareness and acceptance of mental illness. Second, this improved understanding led to greater sympathy for patients with psychiatric illnesses. Lastly, psychiatry continued to shift towards biological explanations for disorders, which helped lift the shroud of pseudoscience and misunderstanding that previously surrounds the field.

## 5. Post-Modern History (1950-Present)

### 5.1. Burn Injury and Depression

The improved excision, grafting and nutritional care of the late 20th century coincided with an increased awareness of the psychological components of burn treatment. A study of 12 patients at the U.S. Army Institute of Surgical Research/Brooke Army Medical Center Burn Center was one of the first to describe the emotional struggles of burn victims [58]. Dr. Alexandra Adler’s detailed observations of the psychiatric complications suffered by the victims of the Cocoanut Grove Fire in 1942 mark one of the earliest efforts within the scientific community to understand the link between post-traumatic stress reactions, depression and burn injury [59].

Despite this early awareness, burn treatment protocols in the 1950s reflect limited progress in our understanding of the relationship between psychological stress and trauma. In The Essentials of Burn Therapy, which was published in 1958, Curtis Artz and Byron Green note that “it is easy to become so engrossed with the problems of fluid therapy and wound care that the patient as a whole is neglected”, concluding that “[to] heal the wounds and end up with a psychological cripple is to fail in the complete treatment of the severely burned patient” [60]. While progress was being made to identify psychological complications of burn trauma, there remained a lack of understanding about disease severity and complexity. Artz and Reiss, in their 1957 book Treatment of Burns, highlighted the value of trust between the burn victim and their care team, suggesting that “each day the physician must sit at the patient’s bedside, if only for a short time, and give assurance of his sincere concern [for them]” [61]. While they acknowledged the need for “emotional rehabilitation”, they placed that duty solely on the shoulders of the primary physician, citing that “[the] need for expert psychiatric assistance is rare” [61]. Crasilneck and collogues argued that hypnosis might aid in the rehabilitation of burn patients, showing that it helped improve food intake, morale and attitude [62]. Though today we may consider these treatments lacking, such early efforts to address the psychological impact of burns were pioneering.

Progress continued into the 1970s as researchers uncovered more links between trauma and complex psychological stress reactions. Andreasen began cataloging normal (anxiety, mild depression, fear, pain) and abnormal (severe depression, delirium) reactions to hospitalization among burn patients, noting that surgeons could provide more effective care if they understood the resulting psychiatric complications [63]. Several studies between 1970 and 1975 found evidence of traumatic post-burn reactions marked by depression, anxiety and shock, and sought to find ways to manage the “emotional reactions” of burn victims [63,64,65]. A major shift in understanding post-trauma psychological reactions occurred with the introduction of PTSD diagnosis in DSM-III during the Vietnam war and the increasing mental health needs of returning veterans [12]. In the same year, a diagnosis of major depressive disorder (MDD) was included in DSM-III, establishing specific diagnostic criteria for unipolar depression. These criteria have been refined in the following revisions of the DSM. According to DSM-V, the current diagnostic criteria for a major depressive episode requires (1) depressed mood and/or (2) loss of interest or pleasure in nearly all activities that (3) lasts at least 2 weeks and marks a change from an individual’s previous level of functioning [66]. At least one of these two symptoms must be accompanied by other common symptom of depression—resulting in 5 symptoms in total. Additional symptoms include changes to appetite or weight, disturbed sleep, fatigue, lack of energy, feelings of worthlessness, hopelessness or guilt, difficulty concentrating or making decisions, psychomotor retardation, agitation or suicidal ideation [66].

In the early 2000s, the medical literature saw an influx of studies investigating the prevalence and risk factors of depression among burn survivors [67]. Reported prevalences on the severity of depressive symptoms and the time at which these symptoms were measured. The prevalence of major depressive symptoms at discharge was reported to be 4% overall, while moderate and mild symptoms were reported to be as high as 26% and 54%, respectively, [16]. Additionally, prevalence has been shown to increase between the first and second year post-discharge, suggesting prevalence rates may be underreported by those measuring short-term prevalence [67]. The evaluation of risk factors for post-burn depression has been marred by similar variability, however, no studies have found burn severity to predict depression [16]. The difficulty in quantifying depression in burn victims is two-fold. First, methodological limitations expose the current literature to a high level of potential bias [16]. Small sample sizes, single institution studies and a lack of standardized measurements of depression in burn survivors has limited the utility of recent research [16,68]. Different studies use different scales and cut-offs to establish diagnoses, few of which have been validated in a populaation of burn injury survivors. The introduction of the Burn Specific Health Scale (BSHS) and Young Adult Burn Questionnaire attempts to provide standardized measurements specific to burn patients, but these scales currently do not have the power to diagnose depression independently. In addition, of the 56 articles on burns and depression reviewed by Wiechman and collegues in their 2016 review, only one measured patients’ symptoms two years post-injury [68]. While a few other authors have attempted more longitudinal studies, the long-term sequelae of burn injury require more extensive investigations [69].

Quantifying depression in burn patients has also been difficult due to the overlap in somatic symptoms. Difficulty sleeping, fatigue and appetite changes that are often indicative of depression are also common sustained adverse effects of burn injuries [68]. This overlap makes the diagnosis of depression convoluted and potentially subjective in burn patients.

Another barrier to diagnosing depression in burn survivors is the high comorbidity rates between PTSD and depression with several overlapping symptoms, including sleep and concentration problems, negative alterations in cognitions and mood, and a lack of interest in previously enjoyable activities [70]. While PTSD may account for many depressive symptoms in the aftermath of trauma, a separate diagnosis may still be given, particularly if there is evidence of prior depressive episodes preceding the burn injury. Regardless of the diagnosis, identification of psychiatric maladjustment following burn injury warrants effective clinical intervention to improve psychosocial adjustment and quality of life post-burn.

While a consensus on the prevalence and risk factors for burn-related depression has not yet been achieved, it would help providers mitigate predicable psychological outcomes. In the meantime, physicians are challenged to treat patients’ symptoms with the limited criteria-based scoring systems they have at their disposal. Fortunately, such assessment tools for depression have been validated in other trauma populations and are appropriate for use in burn populations.

### 5.2. Antidepressants

Tricyclic antidepressants (TCA) first emerged as a treatment for depression in 1951 [36]. TCA, which block the reuptake of serotonin and norepinephrine in presynaptic terminals, eventually became the first pharmacological treatment used in burn patients with depression [71]. Since then, a number of TCA, which include amitriptyline, amoxapine, doxepin, desipramine, nortriptyline, protriptyline and trimipramine, have been approved by the U.S. Food and Drug Administration and used as pharmacotherapy for depression.

Two studies found a response rate of over 80% in burn patients who exhibited stress and depressive disorders. While these observations were promising, there were reasons to be cautious [71,72]. TCA therapies work by inhibiting neuronal uptake of norepinephrine and serotonin, thereby increasing the levels of these neurotransmitters and improving modulation of mood, cognition and anxiety [73]. However, serotonin promotes cellular viability and proliferation of fibroblasts and keratinocytes, which are necessary for proper wound healing [74]. Thus, positive patient response to TCA could not be causally identified, as it is possible that burn victims taking TCA were experiencing improved mood symptoms due to enhanced wound healing from increased serotonin availability, rather than a pharmacological effect on their mood modulation. While this alternate causality is still beneficial from a patient perspective, it is unclear if TCA would benefit patients whose depressive symptoms did not stem from the progress of their wound healing.

Wound healing is not the sole mechanism through which depressive symptoms appear. For instance, children with more adverse childhood experiences were less resilient when facing stressful events and reported more depressive symptoms following burn injury [75]. One analysis even found that certain personality factors, such as high neuroticism, aggression and hostility were more likely to predispose a patient to experiencing depressive symptoms following a burn injury [76]. Therefore, it is necessary to identify therapies for burn-related depression that work independently of wound healing.

Additionally, while TCAs were initially hailed as a solution to depression and widely adopted, their adverse effects quickly led to a need to develop safer therapies. TCAs were found to act as competitive antagonists on alpha-cholinergic, muscarinic and histaminergic receptors [77,78,79]. Perhaps the most alarming side effects of TCA are the cardiovascular complications. Early studies indicate TCA may have arrhythmias leading to QT interval prolongation, ventricular fibrillation and cardiac death [80]. Thus, despite their efficacy, the scientific community has look towards alternative therapies for treating depression since their introduction.

### 5.3. SSRIs

In the late 1980s and early 1990s, a new class of antidepressants emerged—selective serotonin reuptake inhibitors (SSRIs). These included drugs that are commonly used today in the treatment of depression and other mental disorders, including Zoloft^®^ (Sertraline), Prozac^®^ (Fluoxetine) and Paxil^®^ (Paroxetine). These were quickly adopted by the burn community to treat burn patients with depression and several studies validated the efficacy of SSRIs in alleviating or preventing depression and PTSD symptoms in patients following a burn injury [72,81,82].

While these results were promising, there are two reasons for caution. The first reason is the same as previously discussed. SSRIs work by inhibiting the reuptake of serotonin from the synaptic cleft, increasing serotonin levels in the body and confounding the reasoning behind their benefit in burn-related depression.

The second reason for caution is arguably far more important. A significant clinical finding called serotonin syndrome has been identified as a potentially life-threatening condition that consists of mental status changes, neuromuscular hyperactivity and autonomic hyperactivity. In severe cases, patients develop hyperthermia, seizures, renal failure and respiratory failure [83]. Serotonin syndrome is notably caused by therapeutic use of serotonergic drugs alone (such as SSRIs) or by complex drug interactions with serotonergic drugs.

Pain management in burn patients is of utmost importance and often involves multiple drugs from different classes and varying receptor activity. This heightens the risk for severe drug interactions, especially for those on SSRIs. Problems arise when many of the early symptoms of serotonin syndrome are masked by sedation and other adverse effects seen after major burn injuries. Because of this complex interaction and heightened risk in burn victims, SSRIs are not the optimal line of treatment for depression in burn patients.

The unique metabolism of burn patients is also a major consideration for pharmacologic therapy. Burn patients undergo a biphasic metabolic response to injury, with the first 48 h characterized by hypovolemia, fluid retention and low glomerular filtration rate and the second phase characterized by increased blood flow to the kidneys and liver, and exudate leakage [84]. These changing parameters alter the pharmacokinetics of burn patients, often in individualistic ways that requires further research. Thus, TCA and SSRI therapy, as well as future pharmacologic innovation, must consider the metabolic effects of burn injury.

### 5.4. Cognitive Behavioral Therapy

While pharmacological solutions to depression and mood disorders were growing in popularity, a new form of treatment was establishing itself. In 1979, Aaron Beck first described a clear format for using cognitive therapy for depression [85]. His method emphasized the need to focus on conscious thinking and relationships between cognition, emotio and physiology. The premise postulated that by careful examination of behavior and thought processes, emotional reactions due to inaccurate or distorted thinking could be reduced [85]. The method engages the patient by encouraging the development of a coherent personal story for narrative formation. Then, the therapist determines a problem list, defines clear goals for the session, and provides a take-home task following the session [86].

From a behavioral perspective, depression can be conceptualized as lack of positive reinforcement from the environment. To this end, another empirically supported treatment for depression emerged in 1973, called Behavioral Activation (BA), derived from Charles Ferster’s Functional Analysis of Depression [87]. BA focuses on identifying, scheduling, and monitoring pleasurable activities to increase the patient’s contact with sources of positive reinforcement [88]. The goal is to help patients engage or re-engage in previously enjoyable and meaningful activities consistent with a patient’s values, promoting social support, and facilitating a sense of accomplishment.

Cognitive Behavioral Therapy (CBT) typically combines the focus on maladaptive beliefs and cognitions with behavioral interventions to produce a synergistic effect on the patient’s mood [89,90]. As one of the most evidence-based interventions for treating a variety of psychiatric conditions, CBT is used to treat depression, anxiety, panic, substance abuse, eating disorders, insomnia, anger, personality disorders and psychosis. Furthermore, CBT has been widely adopted by the medical community due to its wealth of supportive literature [91,92]. Among burn victims, CBT was first introduced for pain management, but it was not until much later that it was utilized for burn-related depression. CBT holds significant promise for depression treatment in burn patients because it addresses the potential root causes of depression. For instance, data collected for up to five years post-discharge following burn injury found burn injuries induced greater body dissatisfaction, which in turn increased depressive symptoms for up to five years after hospital discharge [69]. Similarly, neuropsychiatric interviews with burn patients have shown that the most accurate predictors of PTSD and depression are the risk of social exclusion and low-body image adjustment [93]. These cognitive-behavioral perceptions influence the patient’s psyche and produce symptoms based on external factors that feel out of their control. CBT, therefore, became a leading candidate for behavioral assessment and positive mindset induced therapy.

While the development of CBT began nearly 40 years ago, the application of such methods to burn patients is a relatively recent concept. In their 2015 pilot study for treating psychosocial consequences of burn injury, Cukor and collogues, utilized a 14-session treatment protocol using cognitive behavioral interventions, such as behavioral activation, modeling, and imaginal exposure. PTSD scores decreased 36%, depression scores decreased by 47%, and patients reported significant improvement in self-image and community reintegration [94]. More recently, it has been found that Safety, Meaning, Activation and Resilience Training (SMART) intervention—which is a 4-session cognitive behavioral therapy-based psychological intervention—reduced stress, posttraumatic and major depressive scores below clinical cutoffs [95].

CBT has been proven to be effective on several occasions, but there remains significant work to be done to prove its efficacy in the long term. These interventions are most promising because unlike TCA and SSRIs, CBT’s approach is far more holistic and targets the origin of the depressive symptoms. As such, we should continue developing therapies that involve the use of behavioral therapy for mood regulation and cognitive improvement.

### 5.5. Interdisciplinary Therapy

Beyond individual psychotherapy, a number of additional services and social changes have gained popularity in recent years to support the emotional and psychological recovery of burn trauma survivors. The first of which is the availability of group therapy. In addition to CBT, group therapy with other survivors affords patients an informal environment to continue to progress in their trauma therapy work while gaining affirmation, empathy, and motivation from peers with shared experiences. Building upon the success of group therapy, some institutions have established so-called “burn camps”. The majority of these programs are geared towards paediatric patients and allow a constructive environment to normalize the the survivor experience and build social confidence. In addition, many behavioral health services encourage survivors to reconnect with prior hobbies or passions. Such extracurricular investments afford social engagement and encourage physical rehabilitation. Emotional support animals have also been advocated after complete wound healing has occurred to provide patients with non-verbal companionship to cope with fluctuating feelings of isolation, anxiety, depression and management of traumatic triggers within the environment. Most importantly, there has been a linguistic change in nomenclature. Patients who were previously referred to as “burn injury victims” are now identified as “burn trauma survivors”. Such a simple change promote a positive idenity, and discardes the shadow of re-traumatization and victimhood.

The Phoenix Society offers resources to burn survivors, their loved ones and healthcare professionals, is an advocate of many of the interventions and resources mentioned. The Phoenix Society also provides new burn survivors a multitude of electronic reference material, as well as access to in-person or virtual group sessions. The availability of such valuable resources has markedly reduced the barriers to engagement and investment in mental health following burn trauma. The impact and normalization of such interventions has not yet been well studied nor has been a standardization of outcome measurement, but given the potential for significantly improvement in depression and stress reactions, established burn centers are increasingly making such resources are available to all burn survivors.

## 6. Conclusions

The last 80 years have seen significant improvement in addressing mood disorders and depression, specifically in burn survivors. (Appendix A: Milestones in the Management of Depression and Stress Reactions in Burn Care). This has been the result of changes in medicine-wide care practices, increased awareness about mental health following traumatic injury and targeted research. Previously, burn patients experiencing features of mood dysregulation were insufficiently managed and in some cases out-right ignored. Today, most specialized burn centers recognize that such psychological symptoms are adverse predictors of recovery and peruse a proactive approach to engage at-risk patients and direct them to the appropriate resources. This review article has sought to record this paradigm shift over the course of time. Certainly a limitation of our work is the lack of documentation in the achient past and pre-modern times. For a number of reasons, ranging from limited medial awareness to non-sophisticated record keeping and social stimitaization, very little is known before the 20th century about the recognition and treatment of trauma-related psychiatric reactions. What little that does exist is heavily weighted with anecdotal assumption, religious appropriations and Galenic-like theories, leaving the central medical information difficult to extract. Although societal awareness about psychiatric reactions to trauma has substantially improved over the last century, decades of social stigmatization and mistreatement of patients has stifled early academic growth.

Although, inherently limited in our ability to include every contribution from the past three millennia given the gaps in record-keeping, nuanced influence of early efforts and the limited specificity to burn-related mental trauma for the majority of advances in psychiatry, this review has endeavored to characterized the changes in management of burn-related depression and stress reactions in a comprehensive manner. Evidence-based research from the past decade fails to appreciate the individual, anecdotal and social movements, which have influenced the framework for post-burn mental health care. This review coalesces the available records as well as the failures and missteps of the past into a comprehensive narrative to explain the modern standard of care.

Despite profound improvement in patient outcomes, many challenges remain. Moving forward, the focus of the burn care community must be on the psychological effects of burn injuries and striving to optimize treatments for survivors. It is not enough to present patients with therapy options months after discharge or to propose treatments that are financially impractical. Instead, we must ensure that therapeutic interventions, such as CBT and SMART, in conjunction with pharmacological solutions are available to patients early in their care. Practical long-term investments like group-therapy and resources like the Phoenix Society need to be routinely provided to all patients deemed to be at risk. Such changes to clinical practice require a greater consensus among providers about the prevalence of burn-related depression and stress reactions, as well as an academic acknowledgement of the clinical value of such investments. Currently, multiple screening tools are utilized across institutions. The lack of homogeneity among such tools, limits the power of their conclusions, which in-turn impairs multi-institutional adoption and routine application. Future research should focus on identifying and influencing the risk factors associated with burn-related depression and stress reactions. Research on such factors could support early and reliable identification of at-risk patients and better characterize the resources that would have a durable influence on outcome metrics and quality of life. Such changes will establish a foundation from which to collect relevant, patient-centered feedback and optimize care practices in a manner that is more robust and nuanced than is currently achievable. Lastly, pharmacologic therapies for burn-related depression must be studied more extensively—in particular the impact of hypermetabolic changes characteristic of large burn injuries on drug pharmacokinetics and clinical efficacy warrants more exploration. Currently, the literature on the pharmacologic management of burn-related depression is challenging to interpret due to the scarcity of high quality data, small cohort sizes and conflicting conclusions among existing studies, which limit clinical applicability and impedes standardization of practice guidelines across burn centers. Change is, however, on the horizon. For example, a number of trial assessing the efficacy of alternative anti-depression therapies such as psychedelics (Ketamine) and non-psychedelics (serotonin-reuptake inhibitors) are currently underway. Such treatments along with combination therapies have the potentially to substantially influence the current standard of care.

Depression and stress reactions among burn trauma survivors are major problems that must be addressed through effective screening and intervention in both the acute and post-acute care settings. The overall goals of treatment should be to accelerate physical and psychological recovery, reduce functional impairment, improve quality of life, and reduce barriers to reintegration into society. This will only be possible when mood disorders and stress reactions in burn patients are normalized as expected outcomes of traumatic stress. Psychologic recovery is an essential component of a comprehensive burn program and the routine application of screening and intervention services is a standard of care required by the American Burn Association for burn center verification. Mental health providers, therefore, should be considered integral members of the interdisciplinary burn team addressing behavioral and psychological barriers to optimize the psychosocial rehabilitation of burn patients.

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
