# Peer review of "A Narrative Review of the History of Burn-Related Depression and Stress Reactions"

_medicina, 2022, doi:10.3390/medicina58101395_

Round 1
Reviewer 1 Report
Here are my comments and suggestions about the article:
(a) The study is interesting; addresses an important a current topic.
(b) In general, I believe that the document would benefit from a revision of the English language by native-speaking English.
(c) The abstract should be rewritten so that it sounds more scientific.
(d) Have the authors considered the theoretical framework or any theories when conducting the research?
(e) The scientific relevance of the study is not stated (What is the gap in the literature that is addressed by this study?)
(f) Limitations of the study, future lines of research should be emphasized.
Author Response
Revisions for “A Narrative Review of the History of Burn-Related Depression and Stress Reactions”:
To begin with, we would like to share our appreciation for the time and effort the reviewer has invested in improving the quality of our manuscript.
The following changes were made to satisfy the recommendations proposed in the comments and suggestions section by the reviewer.
(a) The study is interesting; addresses an important a current topic.
We thank the reviewer for their support of our work.
(b) In general, I believe that the document would benefit from a revision of the English language by native-speaking English.
We appreciate the reviewer’s feedback, but are somewhat confused by the lack of specificity. Given that all the authors are native English speakers, we sought to address this critique by carefully reviewing the manuscript for grammatical and spelling errors. Minor changes were made throughout the manuscript to optimize word-flow, correct spelling or tense, and replace vague or repetitive language with more succinct statements. Specific changes can be identified in the track-changes tab of our revised manuscript. Overall, we believe these changes improved the quality and clarity of our manuscript.
(c) The abstract should be rewritten so that it sounds more scientific.
We appreciate the reviewer’s feedback, but are somewhat challenged in our response by the lack of specificity given that the article is a historical review and the abstract serves to introduce the scope of the manuscript. That said, we have substantially re-written the abstract to remove narrative or repetitive language and replace it with clearer wording acknowledging the contribution of past advances in establishing the modern practice paradigm (lines 10-21).
(d) Have the authors considered the theoretical framework or any theories when conducting the research?
We appreciate the reviewer’s feedback, but are somewhat challenged in our response by the lack of specificity regarding what specific theoretical framework or theories the reviewer is referring to. As the manuscript is a historical review and not a contribution of a new data, it inherently lacks a hypothesis and methodology against which existing theoretical infrastructure would typically be challenged. That said, multiple paragraphs in the introduction and conclusion were revised to provide a clearer understanding of the limited value of past contributions and potential topics that future research could explore to challenge existing theories. In addition, we revised our purpose statement in the introduction (lines 140-144) and emphasized the academic goals our narrative seeks to address (lines 151-153). We expanded upon our review of modern practices by adding a section on interdisciplinary therapy (section 5.5, lines 574-603) to emphasize the importance of different treatments and resources available for burn survivors. Finally, we made substantial changes to the conclusion paragraphs to identify future areas of consideration for research (lines 624-650)
(e) The scientific relevance of the study is not stated (What is the gap in the literature that is addressed by this study?)
We appreciate the reviewer’s feedback. We sought to address the academic significance of our manuscript in a revision of the conclusion paragraph, by describing how our work bridges the gaps in the current literature and identifies future research areas for considerations (lines 606-623).
(f) Limitations of the study, future lines of research should be emphasized.
We appreciate the reviewers feedback. We sought to address the limitations of this work within our revision of the conclusion section by acknowledging the challenges inherent to constructing a comprehensive historical review of a topic for which resources are limited, at times conflicting and in many cases are not burn trauma specific (lines 606-623).
We greatly appreciate the time and effort taken to review our manuscript and hope that these changes meet the expectation of the reviewer and are consistent with the revisions they envisioned.
Reviewer 2 Report
An excellent overview on the history of psychiatry & psychiatric treatment in burns patients/ survivors.
Although not strictly deemed an original research article per se or a meta-analysis ,I believe this paper has merit for publication to raise awareness for burns surgeons in the holistic management of burns injuries.
Author Response
Revisions for “A Narrative Review of the History of Burn-Related Depression and Stress Reactions”:
To begin with, we would like to share our appreciation for the time and effort the reviewer has invested in improving the quality of our manuscript.
Regarding our revisions to the manuscript, as no specific points were identified by the reviewer within the comments and suggestions section, we did not make any specific modifications to the manuscript that warrant highlighting. However, as the reviewer did note that they felt the "English language and style are fine/ minor spell check required", we carefully reviewed the manuscript for grammatical and spelling errors. Minor changes were made throughout the manuscript to optimize word-flow, correct spelling and tense, and replace vague or repetitive language with more succinct statements. Specific changes can be identified in the track-changes tab of our revised manuscript. Overall, we believe these changes improved the quality and clarity of our manuscript.
Thank you for your continued support of our efforts to produce a valuable contribution to the field of burn care.